# Implicit Neural Representation as vectorizer for classification task applied to diverse data structures

**Thibault Malherbe**
Inetum

## Abstract

Implicit neural representations have recently emerged as a promising tool in data science research for their ability to learn complex, high-dimensional functions without requiring explicit equations or hand-crafted features. Here we aim to use these implicit neural representations weights to represent batch of data and use it to classify these batch based only on these weights, without any feature engineering on the raw data. In this study, we demonstrate that this method yields very promising results in data classification of several type of data, such as sound, images, videos or human activities, without any prior knowledge in the relative field.

## 1 Introduction

Implicit neural representations (INRs) have shown great promise in a variety of tasks, including image and shape synthesis, rendering, and inversion. INRs are neural networks that learn to represent a high-dimensional space implicitly without requiring an explicit parametric form for the function.

When it comes to classifying subsets of data using INRs, there are a few approaches that can be taken. One possible approach is to use INRs to learn a representation of the data that is optimized for classification. This can be done by training an INR on a dataset and using the learned weights as extracted features that are then fed into a classifier such as XGBoost or a neural network.

Extracting relevant information from sets of data is a fundamental task in many fields such as machine learning, data science, and engineering. The process of comparing and classifying sets of data often requires a minimum knowledge of the data itself, such as its statistical characteristics

(min, max, average, etc.) of each of their variables. However, this approach can be limiting as it requires the data to be preprocessed in a specific way, and the choice of statistical characteristics may not always capture the most important features of the data.

In this paper, we demonstrate the effectiveness of using INRs for data classification. We show that the weights of a neural network can be used as a vector representation of a structured data set such as sound, images, videos or accelerometer data, and that this representation allows a model such as XGBoost to accurately classify data. Our experiments demonstrate the potential of INRs for several kinds of data and provide insights into their use for other similar applications.

Furthermore, INR are mainly used for data reconstruction, but this is not our goal here. That's why in this paper we will show if there is a direct correlation between the capacity to reconstruct data and the capacity to represent the data and have a good classification score.

## 2 Background and related work

Implicit neural representations (INRs) appear to be a good way to represent signals by continuous functions parameterized by neural networks. It has been used to represent diverses kinds of data such as shape parts (Genova, Cole, Vlasic, et al. 2019; Genova, Cole, Sud, et al. 2019), objects (Park et al. 2019; Michalkiewicz et al. 2019; Atzmon and Lipman 2020; Gropp et al. 2020), or scenes (Sitzmann, Zollhöfer, and Wetzstein 2019; Jiang et al. 2020; Peng et al. 2020; Chabra et al. 2020) but also images (Strümpler et al. 2022; Feng, Jabbireddy, and Varshney 2022), videos (H. Chen et al. 2021; Z. Chen et al. 2022) and audio (Szatkowski et al. 2022; Szatkowski et al. 2023; Lanzendörfer and Wattenhofer 2023).
All papers using implicit neural representations do not use them the same way. There is several applications such as super-resolution, compression or interpolation (Sitzmann, Martel, et al. 2020).
In particular SIREN (Sitzmann, Martel, et al. 2020) appears to be the most used architecture in all cited applications. SIREN architecture demonstrate that it's possible to cor-

Accepted pre-registered proposal at the 1[st] ContinualAI Unconference, 2023, Virtual. Full report to follow. Copyright 2023 by the author(s).

rectly perform classification task using INRs in the process but not directly on the weights of the INR (Xu et al. 2022).

# 3 Hypotheses and proposed method

## 3.1 Implicit neural representation

An implicit neural representation (INR) is a type of neural network architecture that learn to represent objects or scenes in a way that is independent of their explicit geometry or parametric representation. In other words, an INR is a neural network that can learn to represent complex shapes and patterns without explicitly encoding their geometry, topology, or other explicit mathematical descriptions.

An INR can learn to represent any type of data, whatever the dimension. By processing the raw signal data through a NN INR, the resulting vector representation can capture the salient features and patterns in the signal data without requiring explicit knowledge of the underlying signal processing or physics.

The generation of INRs is a consequence of machine learning procedures conducted on a specific data collection. Thus, it is imperative to treat the data intended for categorization as a well-structured dataset. Consequently, we adapt our data manipulation techniques to accommodate one dimension less. This is exemplified in the instance of a video, which can be dissected and interpreted as an image dataset. The neural network architecture typically employed for INR execution is likely a 2DConv neural network. Dimensionality reduction can be leveraged as a strategic advantage, allowing us to approach our data as a more conventional problem. This, in turn, simplifies the architecture of the neural network required for the task.

## 3.2 INR vectorization generalization

Let's consider a classification problem with a dataset E, with n classes and P features such as:
$n \in \mathbb{N}+$,
$P \in \mathbb{N}+$,
E = {A1, A2, ..., AK}, where K is the number of subset that compose E,
$\forall i \in [1, K]$, $Ai \subset E$, with $n(Ai) > 0$ and $n(Ai) \leq n(E)$,
$\forall i, j \in [1, K]$ such as $i \neq j$, $Ai \cap Aj = \emptyset$,
Every subset Ai could be classified in class Ci, with $Ci \in \{C1, C2, ..., Cn\}$.
Now let's take a neural network model M, with any achitecture and k weights. $\forall i \in [1, K]$, M is trained from scratch on Ai with a constant initialization. $\forall i \in [1, K]$, we obtain a vector Wi composed of k values, that are the weights of the model trained on Ai.

Now each Ai subset is vectorized into a Wi vector, that can be classified with any standard classifier.

## 3.3 Functions comparison

If a function represents a set of data, then it is possible to compare these functions directly, and in particular to classify these functions in the context of a classification.

In the context of implicit neural representation, the set of weights of a model trained on a subset of data represents this subset, and thus allows these weight vectors to be classified via a more conventional machine learning algorithm. This set of weight could be named *functa* (Dupont et al. 2022), "*a concise term for INRs that are to be thought of as data*".

According to XGBoost research (T. Chen and Guestrin 2016), XGBoost is a highly effective machine learning model for classification tasks, demonstrating superior performance through its regularized model formalization that effectively controls overfitting. Its parallelizable nature allows it to leverage the capabilities of multi-core computers, enhancing its speed and efficiency. Furthermore, XGBoost is versatile, capable of handling a variety of data types, missing values, and outlier values, and can be applied to both regression and classification tasks, including those involving categorical features. In the case of *functa* classification, the main advantage of XGBoost is that even in scenarios with a high number of features, XGBoost performs well due to its tree-based nature, which is renowned for handling high-dimensional data effectively, because depending to the architecture used to create *functas*, the number of features in the Machine learning processus could explodes.

However, it is important to note that XGBoost, like any machine learning model, can be affected by the "curse of dimensionality" when the feature space becomes exceedingly large, making it challenging for the model to identify patterns.

## 3.4 Reconstruction and classification correlation

The quality of an Implicit Neural Representation is often measured by how well it can reconstruct or generate the data it was trained on. This can be quantified using various metrics depending on the specific task. For instance, in image generation, one might use metrics like Peak Signal-to-Noise Ratio (PSNR), Structural Similarity Index (SSIM), or perceptual loss based on pre-trained networks. For 3D shapes, one might use Chamfer distance or Earth Mover's distance. In the case of graph data, one might use graph-based metrics like Graph Edit Distance or subgraph matching scores. But here we don't really want a good INR but a good *functa* so whatever the type of data processed, the only metric that interest us is the accuracy of our XGBoost classifier.

During our experiment we will demonstrate that the INR quality (capacity of data reconstruction) and the *functa*

quality (capacity of being well classified) are not always correlated, and depending on our objectif, reconstruction or classification, we will not choose the same architecture of neural network.

## 4 Experimental protocol

The use of different types of data in our experiments is of paramount importance. The reason for this is twofold. Firstly, it allows us to evaluate the generalizability of functa across different data modalities. An approach that performs well across diverse datasets is likely to be more robust and versatile. Secondly, different types of data come with their unique challenges and characteristics. For instance, image data might involve dealing with high-dimensional inputs and complex spatial dependencies, while audio data might involve handling temporal dependencies. By testing on different types of data, we can gain insights into how well functa can handle these different challenges, which can guide future research and development.

### 4.1 Datasets

To test the approach and demonstrate its generalizability, we will work on four types of datasets: image, sound, video, and accelerometers.

#### 4.1.1 CIFAR-10

The CIFAR-10 dataset is a widely used collection of images for research in image classification. It is commonly employed in the fields of computer vision and machine learning to evaluate and compare the performance of image classification algorithms. It consists of a total of 60,000 color images divided into 10 different classes, with 6,000 images per class. The classes include common objects such as cars, airplanes, birds, cats, dogs, and more. Each image is 32x32 pixels in size and encoded with three color channels (red, green, blue).

This dataset is interesting for classification research due to several challenges it presents:

- Class complexity: CIFAR-10 classes can be challenging to distinguish due to their visual similarity. For example, images of cats and dogs can be quite similar in terms of shape and color, making classification more difficult.

- Instance variability: Images within each class exhibit significant variability in terms of pose, orientation, scale, brightness, etc. This variability requires robust classification models capable of generalizing well to new instances.

- Dataset size: With 60,000 images, CIFAR-10 provides a sufficient amount of data to train machine

learning models and evaluate their performance meaningfully.

- Image size: The 32x32 pixel images are relatively small compared to other datasets like ImageNet. This makes machine learning models lighter and faster to train, facilitating experimentation and iteration in research.

#### 4.1.2 ESC-50

The ESC-50 (Environmental Sound Classification) dataset is a widely used dataset in research on environmental sound classification. It is specifically designed for the analysis and classification of sounds from everyday environments. It consists of a total of 2,000 audio clips, each lasting 5 seconds. The clips are divided into 50 different categories, representing various sounds such as dog barking, car horns, ocean waves, bird chirping, phone ringing, and more. Each class contains 40 audio examples.

This dataset is interesting for sound classification research for several reasons:

- Class variety: ESC-50 offers a wide variety of environmental sounds from different sources and contexts. This allows researchers to study and develop classification models capable of recognizing and distinguishing a wide range of real-world sounds.

- Classification challenges: Classifying environmental sounds can be complex due to acoustic variations, background noise, overlaps, and other factors. ESC-50 provides a realistic testing environment to evaluate the models' ability to tackle these challenges.

- Dataset size: With 2,000 audio clips, ESC-50 provides a sufficient amount of data to train classification models and evaluate their performance meaningfully.

#### 4.1.3 HMDB-51

The HMDB-51 (Human Motion DataBase) dataset is a widely used dataset in research on human motion classification. It is specifically designed for the analysis and recognition of actions and movements performed by humans. It consists of a total of 6,766 video clips from 51 different action classes. The classes include movements such as walking, running, jumping, smiling, waving, lying down, dancing, and more. Each class contains a variable number of videos, with an average of about 70 videos per class. The videos are captured in diverse contexts, with different individuals, camera angles, lighting conditions, and more.

This dataset is interesting for research in human motion classification for several reasons:

- Action variety: HMDB-51 offers a wide range of human movements and actions, spanning from basic actions like walking and running to more complex actions like dancing and sports. This allows researchers to study and develop classification models capable of recognizing an extensive range of human actions.

- Recognition challenges: Recognizing human actions from videos can be challenging due to variations in poses, clothing, backgrounds, camera angles, and more. HMDB-51 provides a realistic challenge to evaluate the models' ability to identify and classify human movements under diverse conditions.

- Dataset size: With over 6,000 video clips, HMDB-51 provides a significant amount of data to train and evaluate classification models. This enables conducting experiments and statistically robust studies.

### 4.1.4 UCI-HAR

UCI-HAR dataset was developed to help in human activity recognition field (Jorge et al. 2012), which involved 30 volunteers engaging in various daily activities such as sitting, lying down, walking, standing, walking upstairs, and walking downstairs. The authors used a smartphone equipped with an accelerometer and gyroscope to capture tri-axial linear acceleration and angular velocities. The data was sampled at a rate of 50Hz and consisted of nine features. The data was then segmented into fixed-width sliding windows with 50% overlap, resulting in a total of 10,299 samples that have been segmented by user id.

### 4.2 Data Computation

Each dataset have to be processed differently. For image data (CIFAR-10), this involves training an implicit neural representation (INR) for each image. For audio data (ESC-50), this might involve training an INR for each audio clip. For video data (HMDB-51), this might involve training an INR for each video frame or sequence. For sensor data (UCI-HAR), this might involve training an INR for each sensor reading sequence.

There are several ways to create INR on each type of data. In all cited papers, INR to reconstruct image is a fonction that map the coordinate of of pixel to its value. But an image could be seen as a time series, and a LSTM could be trained on it, and, given the N first pixel values, predict the N+1 pixel value.

For each dataset we will test different approaches and discuss the relevance of the INR and the *functa* created this way. It will permit us to see if their is any direct correlation between the INR quality and *functa* quality. Moreover, we will be able to see if certain architectures are more suitable to create good *functa*.

### 4.3 Evaluation metrics

This paper evaluates the effectiveness of *functas* with XG-Boost using various performance metrics, as described below:

Accuracy is defined as the ratio of correctly predicted samples to the total number of samples, where TP denotes true positives, FN denotes false negatives, TN denotes true negatives, and FP denotes false positives.
Accuracy = (TP + TN) / (TP + TN + FP + FN)

A confusion matrix (CM) is a square matrix that provides a complete performance analysis of a classification model. The rows of the CM represent instances of true class labels, while the columns represent predicted class labels. The diagonal elements of the matrix indicate the percentage of points for which the predicted label is equal to the true label.

We will also evaluate the quality of the INR used, with a PSNR wich is most commonly defined via the mean squared error (MSE) between two images.

### 4.4 Baseline comparison

During these experiments we'll need two types of baseline:

- An INR baseline to compare what is a good INR for data reconstruction. That baseline will be classified as *functa* too. To achieve that we will use SIREN as baseline if possible for each type of data.

- A classifier baseline to compare with a more common classifier for the current type of data. We will use 2dConv model as baseline image classifier, a 3Dconv model for videos, 1Dconv model for sound classifier, logistic regression for human activity recognition.

### 4.5 Ablation Studies

To understand the contribution of different components of the functa models, we perform ablation studies, which involve removing or modifying certain components and observing the effect on performance.

### 4.6 Analysis

Finally, we analyze the results, compare the performance of the *functa* models with the baseline models, discuss the results of the ablation studies, and provide insights into why the functa models performed as they did.

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
