# OpenReview forum: "Implicit Neural Representation as vectorizer for classification task applied to diverse data structures"
_continualai.org/CLAI/2023/Unconference_Preregistration_Track — 1st CLAI Unconf_

### Official Review · Reviewer_WGQu · 2023-08-06
**An application that accommodates diverse data structures  and extends the INR application.**

**Clarity:** 2
**Originality:** 1
**Soundness:** 2
**Significance:** 2
**Rating:** 5
**Confidence:** 4

**Review:**

In this paper, the authors proposed the adoption of Implicit Neural Representation (INR) as a feature extractor (vectorizer) for diverse image datasets.


**Strengths:**

- Overall, the use of Implicit Neural Representation sounds solid, as supported by the original INR paper which demonstrates the impressive performance and versatility of the SIREN model across various datasets, including images, videos, and sounds.
- Moreover, the INR model requires fewer training epochs, making it an efficient and powerful choice.
- Building on this promising premise, the authors plan to employ INR as an encoder to generate the vectorizer, thereby expanding the potential applications of INR.

**Weaknesses:**

- The writing is quite rough, particularly in section 3.2.
- The author should address the differences between their approach and the original INR. For example, using the SIREN model as an encoder is not truly novel.



**Questions:**

Since the INR and SIREN are well-defined and have a large following in different areas, it is possible to observe a strong performance by INR. However,I would like to recommend the authors to gain more insight into why INR works well on this specific dataset. For example, identifying where INR outperforms other methods and where it does not. Such analysis would strongly benefit our communities and provide a good starting point for selecting training models for different datasets.

**Protocol:**

- More baseline comparisons, such as with the transformer, should be added.
- The evaluation metrics are not well-defined.
- There are no visualizations for INR. Recommending PCA or t-SNE to visualize the classifications.

---

### Official Review · Reviewer_84x2 · 2023-08-16
**Intriguing idea but out of conference focus**

**Clarity:** 3
**Originality:** 3
**Soundness:** 3
**Significance:** 2
**Rating:** 5
**Confidence:** 1

**Review:**

The paper proposes to represent data with implicit neural representation, using the weights of the neural networks as a vector representation, hypothesizing that it is easier to classify with simple models, such as XGBoost.
The paper proposes an intriguing experimental comparison of multiple datasets, ranging from images to sounds, videos, and accelerometers.
The paper is original, however, it is slightly outside the scope of the conference topics.

**Strengths:**

- The paper is clearly written and all the ideas are clearly explained.
- The paper presents clearly the related works and the baselines.
- The paper has an extensive evaluation protocol since it will report experiments on 4 different kind of data.

**Weaknesses:**

- The paper may be out of the scope of the conference. Using INR for representing data is an intriguing idea which, however, is orthogonal to continual learning. My suggestion would be to link the idea to continual learning (such as demonstrating that INR may be better than other representations for learning on a stream of data).
- Computational complexity: How much computation does this approach require with respect to using neural networks or other types of representation? The paper reports, in Sec. 4.2, that it involves learning an INR for each image, audio, or video frame sequence. How much computation requires learning each INR?

**Questions:**

See weaknesses.

**Protocol:**

The paper reports an extensive evaluation protocol, using four kinds of data: images, sounds, videos, and accelerometers, reporting for each a single dataset. The paper also reports two baselines that will be used to compare the approach.

---

### Official Review · Reviewer_MtfC · 2023-08-20
**Initial Review**

**Clarity:** 2
**Originality:** 2
**Soundness:** 3
**Significance:** 3
**Rating:** 5
**Confidence:** 4

**Review:**

The paper centers around the utilization of Implicit Neural Representations (INR) for acquiring data representations, whose weights are subsequently employed to represent batch of data for models like XGBoost for the purpose of data classification. However, I would like to bring attention to several concerns I have regarding the proposal, as outlined in the comments provided below.

**Strengths:**

- The paper delves into an interesting approach of employing the weights of implicit neural representations for data classification. The comparison drawn between the quality of Implicit Neural Representations (INR) and functa quality adds a layer of interest that extends to the wider research community.
- The background and related work segments are well-described. However, there is room for enhancement in the methodology section, as highlighted in the comment below.
- The experimental evaluation has a comprehensive coverage of diverse datasets, encompassing domains such as images, sounds, videos, and accelerometers.

**Weaknesses:**

- It would be beneficial for the paper to provide a more comprehensive exploration of the novelty and unique contributions of the work. Particularly, a comparative analysis against existing works like SIREN, Xu et al. 2022, and Dupont et al. 2022, which also delve into classification utilizing INRs, could shed light on the distinguishing aspects of your approach.
- The paper's content could benefit from a deeper exploration of the methodology, especially in terms of generating INR representations for different types of data. While section 4.2 briefly touches upon this aspect, allocating more space to elaborate on the methodology itself rather than focusing solely on describing datasets would enhance the paper's overall clarity and depth.
- Additionally, expanding the scope beyond just XGBoost and demonstrating the generalizability of the results to a range of machine learning models, including Multi-Layer Perceptrons (MLPs), would strengthen the paper's impact. Highlighting the cost efficiency of your method could also contribute to its broader relevance.
- There is an opportunity to enhance the clarity of the method by providing more elaborate explanations, particularly regarding the methods employed for different types of data as mentioned above. Addressing typos and improving notations in section 3.2 would also contribute significantly to the paper's overall quality.

**Questions:**

Kindly refer to my comments in the above sections.

**Protocol:**

The evaluation employs a range of datasets and includes comparisons with previous works. Nonetheless, the paper could benefit from providing more comprehensive insights into hyper-parameter tuning, exploring alternative methods beyond XGBoost, and offering a deeper understanding of the reasons behind the potential success of functa models. Considering the diverse array of datasets, each demanding distinct methodologies, the practical achievability within a six-month timeframe raises some uncertainty.

---

### Official Review · Reviewer_e3fa · 2023-08-21
**Not very related with continual learning.**

**Clarity:** 2
**Originality:** 2
**Soundness:** 2
**Significance:** 2
**Rating:** 6
**Confidence:** 2

**Review:**

This study proposes utilizing implicit neural representations' weights to classify batches of data, eliminating the need for raw data feature engineering. The method exhibits strong classification outcomes across various data types (e.g., sound, images, videos, human activities), even without domain-specific prior knowledge.

**Strengths:**

The idea of using implicit neural representations' weights to classify batches of data may be novel .

**Weaknesses:**

The study is not very related with continual learning.

**Questions:**

No question.

**Protocol:**

Good.

---

### Decision · Program_Chairs · 2023-09-12

**Decision:**

Accept

**Comment:**

Dear authors,

Congratulations, your paper has been accepted at the ContinualAI Unconference 2023! We look forward to engaging in further discussions with you and others in the community.

Details will follow shortly regarding camera-ready versions. Please do take the feedback from reviews into account.

Thanks!